# Pomegranate Peel Extract Differently Modulates Gene Expression in Gingiva-Derived Mesenchymal Stromal Cells under Physiological and Inflammatory Conditions

**DOI:** 10.3390/ijms242015407

**Published:** 2023-10-21

**Authors:** Miodrag Čolić, Nataša Miljuš, Jelena Đokić, Marina Bekić, Aleksandra Krivokuća, Sergej Tomić, Dušan Radojević, Marina Radanović, Mile Eraković, Bashkim Ismaili, Ranko Škrbić

**Affiliations:** 1Serbian Academy of Sciences and Arts, 11000 Belgrade, Serbia; 2Medical Faculty Foča, University of East Sarajevo, 73300 Foča, Bosnia and Herzegovina; marinamilinkovic7@gmail.com; 3Faculty of Medicine, University of Banja Luka, 78000 Banja Luka, Bosnia and Herzegovina; natasa.miljus@med.unibl.org (N.M.); aleksandra.krivokuca@med.unibl.org (A.K.); ranko.skrbic@med.unibl.org (R.Š.); 4Institute of Molecular Genetics and Genetic Engineering, University of Belgrade, 11042 Belgrade, Serbia; jelena.djokic@imgge.bg.ac.rs (J.Đ.); dradojevic@imgge.bg.ac.rs (D.R.); 5Institute for the Application of Nuclear Energy, University of Belgrade, 11080 Belgrade, Serbia; marina.bekic@inep.co.rs (M.B.); sergej.tomic@inep.co.rs (S.T.); 6Clinic for Stomatology, Medical Faculty of the Military Medical Academy, University of Defense, 11154 Belgrade, Serbia; erakovic.nino78@gmail.com; 7Faculty of Dental Medicine, International Balkan University, 1000 Skopje, North Macedonia; dr_baki@yahoo.com

**Keywords:** gingiva, mesenchymal stromal cells, pomegranate, periodontitis, gene expression

## Abstract

Pomegranate has shown a favorable effect on gingivitis/periodontitis, but the mechanisms involved are poorly understood. The aim of this study was to test the effect of pomegranate peel extract (PoPEx) on gingiva-derived mesenchymal stromal cells (GMSCs) under physiological and inflammatory conditions. GMSC lines from healthy (H) and periodontitis (P) gingiva (*n* = 3 of each) were established. The lines were treated with two non-toxic concentrations of PoPEX (low—10; high—40 µg/mL), with or without additional lipopolysaccharide (LPS) stimulation. Twenty-four genes in GMSCs involved in different functions were examined using real-time polymerase chain reaction (RT-PCR). PoPEx (mostly at higher concentrations) inhibited the basal expression of *IL-6*, *MCP-1*, *GRO-α*, *RANTES*, *IP-10*, *HIF-1α*, *SDF-1*, and *HGF* but increased the expression of *IL-8*, *TLR3*, *TGF-β*, *TGF-β/LAP* ratio, *IDO-1*, and *IGFB4* genes in H-GMSCs. PoPEx increased *IL-6*, *RANTES*, *MMP3*, and *BMP2* but inhibited *TLR2* and *GRO-α* gene expression in P-GMSCs. LPS upregulated genes for proinflammatory cytokines and chemokines, tissue regeneration/repair (*MMP3*, *IGFBP4*, *HGF*), and immunomodulation (*IP-10*, *RANTES*, *IDO-1*, *TLR3*, *COX-2*), more strongly in P-GMSCs. PoPEx also potentiated most genes’ expression in LPS-stimulated P-GMSCs, including upregulation of osteoblastic genes (*RUNX2*, *BMP2*, *COL1A1*, and *OPG*), simultaneously inhibiting cell proliferation. In conclusion, the modulatory effects of PoPEx on gene expression in GMSCs are complex and dependent on applied concentrations, GMSC type, and LPS stimulation. Generally, the effect is more pronounced in inflammation-simulating conditions.

## 1. Introduction

Chronic periodontitis is a frequent form of periodontal disease and the most common oral health problem, characterized by inflammation and infection of the tissues surrounding the teeth. It is a progressive pathological process that affects the supporting structures of the teeth, including the gingiva, periodontal ligaments, and alveolar bone [1]. Its has been estimated to have a 5–15% prevalence among the general adult population [2]. Periodontitis occurs when the bacteria in dental plaque build up on the teeth and gingiva, leading to an immune response from the host. Over time, this immune response, which was ineffective in removing pathogens, can cause damage to the gingiva and supporting structures, leading to the formation of periodontal pockets, gingival recession, and bone loss [3]. Several factors can contribute to the development of chronic periodontitis, including poor oral hygiene, smoking, genetic predisposition, stress, certain systemic diseases (especially diabetes), hormonal changes, and certain drugs [3,4,5].

Several species of bacteria are associated with the development and progression of chronic periodontitis. Some of the most common bacteria implicated in this condition include *Porphyromonas gingivalis*, *Fusobacterium nucleatum*, *Tannerella forsythia*, *Treponema denticola*, *Aggregatibacter actinomycetemcomitans*, *Filifactor alocis*, and *Prevotella intermedia* [3,6]. In this context, oral-health-associated commensals protect the development and/or progression of periodontitis by inhibiting the growth of periodontitis-associated pathogens [3,7].

Except for bacterial products such as proteolytic enzymes, toxins, and lipopolysaccharide (LPS), immune cells are important players in chronic periodontitis, including keratinocytes, neutrophil granulocytes, mast cells, monocyte/macrophages, dendritic cells, and T and B cells [8,9,10]. The immune cells are involved in the inflammatory response, tissue destruction, and the restriction of inflammatory/destructive processes in the periodontium [11,12].

One cell type whose role in the pathogenesis of chronic periodontitis is not elucidated is the gingiva-derived mesenchymal stromal cell (GMSC). GMSCs are a type of mesenchymal stem cells that can be isolated from gingival tissue. They share many similarities with other dental MSCs, such as fibroblastoid morphology, adherence to plastic substrates, differentiation potential to other types of mesenchymal cells, and the expression of characteristic markers, including those found on pericytes [13,14]. However, GMSCs possess some characteristics different from the rest of dental MSCs, such as easy isolation, absence of spontaneous differentiation, additional neurogenic and epithelial differentiation potential due to gingival origin from the neural crest, and morphological, phenotypical, and telomerase stability after long-term cultures [14,15]. Some potential roles of GMSCs during periodontitis include hard and soft tissue repair, immunomodulatory properties, angiogenesis, and antibacterial effects [14,16]. However, it remains to be answered whether, at a certain stage of periodontitis development, GMSC could promote an inflammatory response that is needed to eradicate bacteria. In this context, the response of GMSCs to LPS is important because this component of the outer membrane of Gram-negative bacteria is known to elicit a strong immune response and trigger inflammation in the periodontal tissues [17].

The primary goal of periodontal therapy is to remove pathogenic biofilms and suppress inflammation. This strategy involves active anti-infective treatment, often combined with surgery to eliminate residual pockets [3,18]. Anti-infective treatment includes the application of systemic antibiotics, antiseptic mouthwashes, local drug delivery of antiseptics and antibiotics, and immunomodulating agents. However, plaque control remains the primary preventive measure [19]. In this context, searching for natural products (phytochemicals) seems to be a good approach because many bacteria causing periodontitis show antibiotic resistance. Plant natural products extracted from *Curcuma zedoaria*, *Calendula officinalis*, *Acacia arabica*, *Azadirachta indica*, *Curcuma longa*, *Cymbopogam*, *Camellia sinensis*, *Ocimum sanctum*, or *Aloe vera* have been used to treat many oral diseases [20,21]. Except for their antiseptic and antifungal properties, these herbal extracts possess antioxidant, anti-inflammatory, and wound-healing properties [21]. In a recent systemic review, Chatzopoulos et al. have shown that herbal products used as adjuncts to scaling and root planing or supragingival debridement led to superior clinical outcomes in comparison with placebo or no adjuncts [21]. Two papers showed similar beneficial effects of *Calendula officinalis* mouthwash [22] or *Aloe vera* gel [23]. To prevent gingivitis, toothpastes containing naturally occurring ingredients, including herbal essential oils, are recommended due to their safety, biocompatibility, and oral-health-promoting ability [24].

Different components from *Punica granatum* (pomegranate), such as peel, seed, and juice, are very rich in various phytochemicals, including ellagitannins, gallotanins, flavonoids, organic acids, and other polyphenols. These compounds possess antioxidant and immunomodulatory properties and have been associated with anti-inflammatory, anticancer, antidegenerative, skin-regenerative, neuroprotective, and cardiovascular health benefits. In addition, pomegranate extract has been shown to possess bactericidal, antifungal, antiviral, and antihelminthic effects [25,26,27]. Numerous studies have shown the beneficial effect of pomegranate extract in various oral diseases and gingivitis/periodontitis [20]. For example, a 10% pomegranate gel applied topically efficiently reduced recurrent aphthous stomatitis and improved ulcer healing. A similar 80% gel had anesthetic effects in the oral cavity. Another pomegranate gel efficiently treated candidiasis-associated denture stomatitis [20,28,29,30]. The pomegranate gels showed a potent anti-gingivitis effect, including acute ulcerative gingivitis and a significant reduction in plaque scores when used as an adjunct nonsurgical therapy [31,32,33]. The positive effects could be explained by the potent antimicrobial activity of the components from the pomegranate extract, their protective effect on oxidative stress, and their ability to suppress inflammation [20].

Considering the facts presented so far, including the important role of GMSCs in physiological and pathological conditions, the question arises regarding how pomegranate peel extract (PoPEx) affects GMSCs because such data are missing in the literature. In this context, a paper showed that kiwifruit extract enhanced the proliferation and migration of human gingival fibroblasts and promoted angiogenesis [34]. In addition, a pomegranate extract and punicallagin exerted antioxidant properties on human gingival fibroblasts but inhibited their proliferation at higher concentrations [35]. We hypothesized that PoPEx differently modulates gene expression in GMSCs established from healthy (H) and periodontitis-affected gingiva (P) based on our previous findings that many functional properties of these lines are different inflammation [15]. Therefore, the main goal of this study was to investigate and compare the effect of different concentrations of PoPEx on the proliferation of H-and P-GMSCs and the expression of genes in these lines involved in the processes of inflammation, immunomodulation, tissue damage, and remodeling, proliferation, senescence, and osteoblastogenesis, under physiological and pathological conditions. To further mimic the processes that occur during periodontitis, the cell lines were additionally stimulated with LPS.

## 2. Results

### 2.1. Establishment and Basic Characterization of GMSC Lines

Gingival samples were obtained from three healthy donors and three donors with periodontitis. The histological analysis confirmed the differences between these groups based on the presence or absence of abundant cellular infiltrate composed of polymorphonuclear and mononuclear cells (lymphocytes, plasma cells, and macrophages) in gingival tissue sections. These changes were not present in healthy gingiva (Appendix A).

The lines exhibited typical fibroblastoid morphology, adherence to plastic, and differentiation capability to osteoblasts, chondroblasts, and adipocytes (Table 1 and Appendix A). They expressed several markers characteristic of MSCs (Table 1 and Appendix A). P-GMSCs showed a higher capability to differentiate towards osteoblasts compared to H-GMSCs. The differences between H- and P-GMSC lines to differentiate into chondroblasts (moderate potential) and adipocytes (low potential) were insignificant. CD90, CD73, CD44, CD73, CD105, and CD166 were expressed on more than 95% of both H-GMSC and P-GMSC lines, CD146, was present on 25–50% of GMSCs, whereas the lowest positivity (10–20%) was seen with antibodies to STRO-1, SSEA4, and pericyte antigens (PDGF-R and NG2). Less than 2% of both H-GMSCs and P-GMSCs expressed hematopoietic cell markers (CD45 and CD34), a myelomonocytic marker (CD14), and T/B-cell markers (CD3/CD19). Although the proportions of cells expressing CD146 and markers of stem cells/pericytes were slightly higher in P-GMSC, the differences were not statistically significant.

One of these monoclonal antibodies (mAbs) (CD146) was used to stain gingival sections. Figure 1a shows positivity associated with small blood vessels. To better analyze these structures, confocal microscopy was performed. The staining of tissue sections with anti-CD146 mAb and anti-CD31 mAb (an endothelial marker) identified CD146 + CD31- pericytes (Figure 1b).

Cumulatively, our results showed that these GMSC lines are very similar to the lines that we had previously established [15] and, as such, were used for the current experiments.

### 2.2. Cytotoxicity and Anti-Proliferative Activity of GMSC Lines

At first, we studied the cytotoxicity of PoPEx on GMSC lines. The results presented in Figure 2A show that both H- and P-GMSCs responded similarly to PoPEx, which decreased cell viability from 60–100 µg/mL in a dose-dependent manner.

Based on these findings, we used lower (10 µg/mL) and higher non-cytotoxic concentrations (40 µg/mL) of PoPEx for further experiments. Figure 2B shows that the treatment with LPS significantly inhibited cell proliferation of both types of cell lines (*p* < 0.05) without inducing cytotoxicity. Higher concentrations of PoPEx had an additional inhibitory effect on the proliferation of LPS-stimulated P-GMSCs (*p* < 0.05).

### 2.3. Effect of PoPEx on the Expression of Cytokine/Chemokine Genes Associated with Inflammation in Control and LPS-Stimulated GMSCs

Control and LPS-stimulated H- and P-GMSCs were incubated with low (10 µg/mL) or high (40 µg/mL) concentrations of PoPEx for 24 h, as described in the Materials and Methods Section. PoPEx showed significant modulation of the most investigated genes. As a rule, higher concentrations had a stronger modulatory effect. When the effect of lower concentrations was pronounced, these results were emphasized.

The expression of cytokine/chemokine genes related to inflammation, including Interleukin-6 (IL-6), IL-8, monocyte chemoattractant protein (MCP)-1, and growth-related oncogene (GRO)-α, was evaluated using qPCR (Figure 3).

The expression of IL-6 mRNA was slightly down- and upregulated by higher concentrations of PoPEx in H- and P-GMSCs, respectively (*p* < 0.05). LPS stimulated IL-6 expression about three times in H-GMSCs (*p* < 0.05) and about 30 times in P-GMSCs (*p* < 0.01), compared to the basal level, and its expression was additionally augmented by PoPEx. Lower concentrations had a much stronger upregulating effect, especially in LPS-stimulated P-GMSC cultures (*p* < 0.005).

IL-8 gene expression was almost non-detectable in both lines at the baseline level, and its expression was increased about 70 (H-GMSCs) and 700 times (P-GMSCs) after LPS stimulation (*p* < 0.005). PoPEx, which had a slight stimulatory effect on H-GMSCs (*p* < 0.05), and significantly augmented and inhibited IL-8 mRNA expression in LPS-stimulated P- and H-GMSC lines, respectively (*p* < 0.01).

PoPEx inhibited basal expression of MCP-1 in H-GMSCs (*p* < 0.05). In contrast, LPS stimulated MCP-1 mRNA expression in P-GMSCs (*p* < 0.05). PoPEx augmented MCP-1 expression in LPS-stimulated cultures, and the upregulating effect was stronger in P-GMSCs (*p* < 0.001) than in H-GMSCs (*p* < 0.05).

PoPEx inhibited basal gene expression of GRO-α in both lines (*p* < 0.05). LPS upregulated GRO-α in P-GMSCs about 50 times (*p* < 0.005) but had no significant effect in H-GMSCs. PoPEx had an opposite dose-dependent effect in LPS-stimulated cultures (stimulatory in H-GMSCs, inhibitory in P-GMSCs) (*p* < 0.05).

### 2.4. Effect of PoPEx on the Expression of Toll-like Receptors, Nuclear Factor Kappa-B 1, and Cyclooxygenase-2 Genes in Control and LPS-Stimulated GMSCs

The group of genes that includes genes such as *Toll-Like Receptors* (*TLR*) 2, 3, and 4, transcription molecule *Nuclear Factor Kappa B 1* (*NFKB1*), and enzyme *Cyclooxygenase* (*COX*)*-2* is involved in the inflammatory pathway. The effects of PoPEx on their expression in control and LPS-stimulated GMSCs are presented in Figure 4.

*TLR2* gene expression was non-detectable in control, LPS-stimulated or PoPEx-treated H-GMSCs. The expression of TLR2 mRNA in LPS-stimulated P-GMSCs was statistically significantly increased (about ten times) compared to the control P-GMSCs (*p* < 0.005). PoPEx completely downregulated the TLR2 mRNA expression by both non-treated and LPS-treated P-GMSCs, in a dose-dependent manner (*p* < 0.05 and *p* < 0.005, respectively).

The expression of TLR3 mRNA by H-GMSCs was almost non-detectable in contrast to P-GMSCs. However, PoPEx more strongly upregulated TLR3 expression in H-GMSCs (*p* < 0.005) compared to P-GMSCs (*p* < 0.05). The stimulatory effect of LPS on P-GMSCs was about ten times higher than H-GMSCs (*p* < 0.005). A lower concentration of PoPEx increased TLR3 in LPS-stimulated P-GMSCs (*p* < 0.005), in contrast to higher concentrations which were inhibitory (*p* < 0.05). A similar trend of modulation, but less strong, was observed with PoPEx on LPS-stimulated H-GMSCs (*p* < 0.05).

LPS inhibited basal expression of TLR4 mRNA expression in H-GMSCs (*p* < 0.05). TLR4 was not expressed in P-GMSCs and was not significantly modulated by LPS. PoPEx treatment did not significantly modulate TLR4 in either control or LPS-stimulated GMSC lines.

The expression of the *NFKB1* gene was almost non-detectable in both H- and P-GMSCs and was not modulated by PoPEx. LPS slightly upregulated NFKB1 mRNA in H-GMSCs (*p* < 0.05) but not in P-GMSCs. PoPEx did not significantly affect LPS-stimulated H-GMSCs, but augmented NFKB1 expression in LPS-stimulated P-GMSCs, in a dose-dependent manner, by about 40 and 150 times, respectively (*p* < 0.005).

PoPEx did not significantly change COX-2 mRNA in both lines. LPS stimulated COX-2 expression about 10 times in H-GMSCs and about 40 times in P-GMSCs (*p* < 0.01). Lower concentrations of PoPEx upregulated COX-2 expression in LPS-stimulated P-GMSC lines (*p* < 0.005). In contrast, higher concentrations downregulated COX-2 in LPS-stimulated H-GMSCs (*p* < 0.05) but additionally upregulated its expression in LPS-stimulated P-GMSCs (*p* < 0.005).

### 2.5. Effect of PoPEx on the Expression of Genes Associated with Immunomodulation in Control and LPS-Stimulated GMSCs

Five genes associated with immunomodulation have been studied, and the results are presented in Figure 5.

Indoleamine 2, 3-Dioxygenase 1 (IDO-1) mRNA was almost non-detectable in both H- and P-GMSCs but was significantly increased upon LPS stimulation (*p* < 0.005). Both concentrations of PoPEX additionally upregulated IDO-1 in LPS-stimulated P-GMSCs (*p* < 0.005 and *p* < 0.01, respectively), as did lower concentrations in LPS-stimulated H-GMSCs (*p* < 0.05). However, higher concentrations of PoPEx inhibited IDO-1 in LPS-stimulated H-GMSC cultures (*p* < 0.05).

PoPEx stimulated transforming growth factor (TGF)-β mRNA expression in control H-GMSCs. LPS did not significantly modulate TGF-β mRNA expression in both lines. PoPEx upregulated TGF-β mRNA in LPS-stimulated P-GMSCs (*p* < 0.05) but decreased its expression in LPS-stimulated H-GMSCs (*p* < 0.05). Latency-associated peptide (LAP) was almost non-detectable in P-GMSCs. PoPEx downregulated LAP expression in H-GMSCs (*p* < 0.05). LPS did not significantly change LAP expression in both lines, and PoPEx did not additionally modulate its expression.

Interferon (IFN)-γ-induced protein 10 kDa (IP-10) mRNA was non-detectable in P-GMSCs. PoPEx completely suppressed its expression in both lines. LPS significantly upregulated IP-10 in P-GMSCs about 150 times (*p* < 0.005) but had no effect in H-GMSCs. PoPEx had little augmenting effect in LPS-stimulated H-GMSCs (*p* < 0.05), in contrast to the additional upregulation of IP-10 in LPS-treated P-GMSCs (*p* < 0.005).

Regulated upon activation, normal T-cell expressed and secreted (RANTES) mRNA expression was completely downregulated in H-GMSCs by PoPEx (*p* < 0.05). However, it was significantly increased by higher extract concentrations about 60 times in P-GMSCs (*p* < 0.005). LPS upregulated RANTES in H-GMSCs more strongly (*p* < 0.005) in comparison with P-GMSCs (*p* < 0.01). PoPEX upregulated RANTES in LPS-stimulated P-GMSCs, in a dose-dependent manner (*p* < 0.005), and almost completely inhibited its expression in LPS-stimulated H-GMSCs (*p* < 0.005).

### 2.6. Effect of PoPEx on the Expression of Genes Associated with Tissue Regeneration/Repair in Control and LPS-Stimulated GMSCs

Six genes associated with tissue regeneration/repair in GMSCs were studied, and the results are presented in Figure 6.

PoPEx inhibited the basal expression of stromal cell-derived factor (SDF)-1 and hepatocyte growth factor (HGF) mRNA in H-GMSCs (*p* < 0.05). Both genes were almost non-detectable in P-GMSCs, nor did PoPEx significantly modulate their expression. LPS inhibited SDF-1 in H-GMSC and augmented HGF in P-GMSCs (*p* < 0.05). PoPEx did not modulate SDF-1 expression in both LPS-stimulated GMSC lines but additionally upregulated HGF mRNA expression in LPS-stimulated P-GMSCs, in a dose-dependent manner for up to three to six times (*p* < 0.05 and *p* < 0.01, respectively).

Hypoxia-inducible factor (HIF)-1α mRNA was hardly detectable in P-GMSCs, and LPS did not significantly modify its expression. Lower concentrations of PoPEx inhibited basal expression of HIF-1α in control H-GMSCs but potentiated its expression in LPS-stimulated P-GMSCs (*p* < 0.05). Similarly, higher concentrations of PoPEx upregulated HIF-1α mRNA in LPS-stimulated P-GMSCs.

Higher concentrations of PoPEx upregulated insulin-like growth factor binding protein 4 (IGFBP4) mRNA expression in H-GMSCs, as did LPS in P-GMSCs (*p* < 0.05). PoPEx augmented IGFBP4 expression in LPS-stimulated P-GMSCs, in a dose-dependent manner (*p* < 0.005). A much lower stimulatory effect was seen with lower concentrations of the extract in LPS-stimulated H-GMSCs (*p* < 0.05).

Matrix metalloproteinase (MMP)-3 mRNA was identified at baseline levels in both GMSC lines. Its inhibitor, tissue inhibitor of metalloproteinase (TIMP)-2, was almost undetected in P-GMSCs. Higher concentrations of PoPEx stimulated MMP-3 expression in P-GMSCs (*p* < 0.05). LPS inhibited TIMP-2 mRNA in H-GMSCs (*p* < 0.05) but significantly increased MMP-3 expression about seven times in H-GMSCs and more than 100 times in P-GMSCs (*p* < 0.005). PoPEx treatment downregulated the LPS-induced expression of MMP-3 in H-GMSCs (*p* < 0.05) but potentiated its expression in P-GMSCs (*p* < 0.01 and *p* < 0.05, respectively). The extract had only a slight stimulatory effect on TIMP-2 in LPS-stimulated P-GMSCs (*p* < 0.05). As a result, the MMP-3/TIMP-2 ratio was decreased from about 260 in LPS-stimulated P-GMSCs to about 70 after treatment of these cultures with PoPEx (40 µg/mL).

### 2.7. Effect of PoPEx on the Expression of Genes Associated with Osteoblastic Differentiation in Control and LPS-Stimulated GMSCs

Three genes involved in the early phase of osteoblastogenesis and a gene inhibiting osteoclastogenesis were studied in GMSC lines, and the results are presented in Figure 7.

Runt-related transcription factor 2 (RUNX2) mRNA was almost non-detectable in H-GMSCs or significantly modulated either by PoPEx or LPS treatment. LPS significantly upregulated (about 8 times) RUNX2 mRNA expression in P-GMSCs (*p* < 0.01), and the effect was significantly augmented by PoPEx (about 2.5 times, compared to LPS treatment) (*p* < 0.01).

LPS inhibited the basal expression of bone morphogenetic protein 2 (BMP2) mRNA in H-GMSCs (*p* < 0.05). Higher concentrations of PoPEx upregulated BMP2 mRNA expression in P-GMSCs (*p* < 0.05), as did lower concentrations of the extract in LPS-stimulated P-GMSCs (*p* < 0.005).

LPS inhibited the basal expression of Collagen, Type I, Alpha 1 (COL1A1) in H-GMSC lines (*p* < 0.05), in contrast to PoPEx, which was ineffective. PoPEx and LPS did not modulate the basal expression of COL1A1 in P-GMSCs when applied separately. However, higher concentrations of PoPEx significantly increased its expression in LPS-stimulated P-GMSCs (*p* < 0.05).

Lower concentrations of PoPEx and LPS applied individually inhibited the basal expression of osteoprotegerin (OPG) mRNA in H-GMSCs (*p* < 0.05). However, lower concentrations of PoPEx increased its expression in LPS-stimulated H-GMSCs. In P-GMSC cultures, PoPEx increased OPG mRNA expression in LPS-stimulated P-GMSC cultures (*p* < 0.05 and *p* < 0.01, respectively).

## 3. Discussion

In this study, we examined the impact of PoPEx on the cytotoxicity, proliferative capacity, and expression of 24 genes associated with various functions of GMSCs. Two GMSC lines were established from healthy gingiva and periodontitis-affected gingiva using the methodology we previously employed [15]. Additionally, one line from each gingival tissue was used from our previous experiment’s cell bank. Both H-GMSCs and P-GMSCs met the criteria necessary for a cell population to possess MSC characteristics, including a fibroblast-like appearance, adherence to substrates, colony-forming ability, characteristic phenotypic profile, and the capacity for differentiation into three mesenchymal cell lineages [36].

A subpopulation of our lines retained the expression of stem cell markers (STRO-1 and SSEA4) [37] and pericyte markers (PDGFR and NG2) even after prolonged passaging (up to eight passages). Through the use of high-affinity antibodies targeting the CD146 molecule (expressed on endothelial cells and pericytes) in combination with an antibody against the CD31 molecule (an endothelial marker) labeled with fluorescent markers suitable for confocal microscopy, we demonstrated in this study that CD146+ GMSCs originate from pericytes rather than endothelial cells. The network of small blood vessels was particularly prominent in the lamina propria of the gingiva affected by periodontal disease. Our findings also support the hypothesis that pericytes are the predominant—if not the sole—in vivo source of tissue in MSCs [38].

Similarly to the findings of our previous study [15] and studies conducted by other researchers [14], we demonstrated that the differentiation potential of GMSCs into adipocytes and chondroblasts is relatively low, while it is high towards differentiation into osteoblasts. The higher differentiation potential of P-GMSCs into osteoblasts compared to H-GMSCs is in line with the results of other authors [39]. However, it contradicts some other published data [40]. The reason for this disparity may be attributed to differences in tissue selection and quality, isolation, and propagation procedures of MSCs, as well as general conditions of their cultivation.

The cytotoxicity of PoPEx in this study was similar to the cytotoxicity observed in human peripheral blood mononuclear cells (PBMC) that we showed previously [27]. However, our results contradict previous results about the relative resistance of normal cells to cytotoxicity by PoPEx or its polyphenols, in contrast to the sensitivity of cancer cells [41]. The toxic effect of different plant derivatives in cell cultures is a well-known phenomenon, and according to publications, it critically depends upon the dose and cells (cell lines) used [42]. Therefore, testing cytotoxicity before using biologically active plant (food) extracts is necessary to find optimal concentrations for in vitro studies. Although we showed that lower and higher concentrations showed different, sometimes opposite effects, generally, higher concentrations had stronger modulatory activity.

The reason for investigating the effect of PoPEx in GMSCs culture has already been emphasized in the Introduction. So far, pomegranate extract has been studied in cases of gingivitis/periodontitis, showing positive effects [20,31,33]. On the other hand, our previous results demonstrated that GMSCs derived from periodontitis-affected gingiva retain various characteristics of the proinflammatory microenvironment from which they were isolated [15]. Therefore, comparing the effects of PoPEx on GMSCs from healthy and diseased gingiva can significantly contribute to understanding the mechanisms of action of biologically active components from the extract while shedding light on the role of GMSCs in physiological and pathological conditions. The design of this study also included LPS due to its proven role, as an endotoxin from periodontal microorganisms, in stimulating inflammatory processes in periodontitis, including its multiple effects on MSCs [43]. Thus, the stimulation of H-GMSCs with LPS imitates the response of these cells in the initial phase of periodontitis. On the other hand, the stimulation of P-GMSCs with LPS may represent the response of GMSCs during the exacerbation of chronic periodontitis.

Bearing in mind that the biological effects of dietary polyphenols are dose-dependent [44,45], we tested two different concentrations (lower one—10 µg/mL—which was the lowest effective concentration in our preliminary experiments; higher one—40 µg/mL—which is a maximal non-cytotoxic concentration). However, to our surprise, PoPEx exerted different effects on particular gene expression depending on the type of GMSCs and the applied concentration of the extract. Generally, PoPEx (mostly at higher concentrations) inhibited the basal expression of most proinflammatory genes (*IL-6*, *MCP-1*, *GRO-α*), genes involved in immunomodulation (*RANTES*, *IP-10*), and genes involved in tissue regeneration/repair (*HIF-1α*, *SDF-1*, *HGF*) in H-GMSCs. However, PoPEx increased the expression of *IL-8*, *TLR3*, *TGF-β*, *TGF-β/LAP* ratio, *IDO-1*, and *IGFB4.*

IL-6 is the dominant proinflammatory mediator in chronic periodontitis, affecting various functions of both specific and nonspecific immune cells. However, under certain circumstances, it can have anti-inflammatory properties [46]. GRO-α and IL-8 are potent chemoattractants for neutrophils [47]. MCP-1 is a chemoattractant for monocytes/ macrophages, while RANTES is a T-cell chemokine [48]. IP-10 (CXCL10) chemokine, which is produced in response to interferon (IFN)-γ, induces chemotaxis, especially inflammatory T helper 1 (Th1) cells into inflamed gingival tissue, stimulates apoptosis and cell proliferation (upon binding to CXCR3A), but inhibited angiogenesis or cell proliferation (upon binding to CXCR3B) [49,50]. These results suggest that under physiological conditions PoPEx tends to suppress most genes in GMSCs involved in inflammation, which agrees with many results published on other cells [51,52,53].

In contrast, upregulation of *TLR-3*, *TGF-β*, *TGF-β/LAP* ratio, and *IDO-1* genes suggest that PoPEx enhanced immunosuppressive properties of H-GMSCs, a new phenomenon not published up to now. Increased expression of TLR3 mRNA, a dsRNA ligand, is associated with a greater ability to direct MSCs toward the immunosuppressive MSC2 phenotype [54]. On the other hand, IDO-1, the enzyme catalyzing the degradation of L-tryptophan into L-kynurenine [55], represents a key target of immunosuppressive mechanisms. TGF-β is one of the main immunosuppressive cytokines [56]. However, it controls proliferation, cell differentiation, apoptosis, angiogenesis, and different immune-mediated mechanisms. Therefore, the role of secreted TGF-β, which is also a product of MSCs [57], is very complex. In this context, the secreted TGF-β is bound to a latent complex consisting of LAP and latent TGF-β-binding protein (LTBP). LAP maintains the latency of TGF-β, while LTBP converts the latent form of TGF-β into the active form [58]. Therefore, an increase in the TGF-β/LAP ratio by PoPEx in H-GMSCs suggests that biologically active components from the extract promote TGF-β activity and thus additionally increase the immunomodulatory properties of these cells. GMSCs, like other MSCs, are also key players in tissue proliferation and regeneration, including regeneration/remodeling of inflamed periodontal tissue [14,16,59].

HIF-1α induces the transcription of many genes involved in increasing oxygen delivery in hypoxic tissues, accumulation of reactive oxygen species (ROS), MSC migration, inflammation, and tissue repair [60]. It upregulates SDF-1 (CXCL12 chemokine). SDF-1 promotes the recruitment and proliferation of MSCs, increases their osteogenic differentiation [61], stimulates angiogenesis [62], and down-modulates inflammation [63]. HGF is a multifunctional paracrine biomolecule that plays a role in tissue regeneration [64] and cell proliferation [65]. It has been shown that MSCs secrete HGF in the presence of inflammatory stimuli [66]. The inhibition of *HIF-1*, *SDF-1*, and *HGF* genes by PoPEx in H-GMSCs aligns with the concept of the anti-inflammatory effect and suppressive action of this extract on the processes of cell proliferation and regeneration of healthy gingival tissue. IGFBP4, an inhibitor of IGF, plays an important role in many cellular processes in MSCs, including cell growth, differentiation, bone metabolism, and cellular senescence [67,68]. In this context, increased expression of the *IGFBP4* gene in PoPEx-treated H-GMSCs may be relevant for inhibition of GMSC proliferation, differentiation, and survival [67,69], and/or modulation of T regulatory cells (Tregs), key players in immunomodulation [70].

In contrast to H-GMSCs, PoPEx upregulated *IL-6*, *RANTES*, and *MMP3* in P-GMSCs, did not modify *IL-8*, *MCP-1*, and most genes involved in immunomodulation and tissue repair, but inhibited *TLR2* and *GRO-α*. These results suggest that GMSCs established from an inflammatory microenvironment (periodontitis) respond differently to PoPEx. In fact, P-GMSCs somehow retain their functional characteristics in vivo. It can be postulated that in the inflammatory microenvironment during periodontitis, their proinflammatory role is desirable as a defense mechanism against periopathogenic microorganisms. The effect of PoPEx could be additionally supportive in that way. However, it is not clear the significance of increased expression of *MMP-3*. MMP-3, as a member of the MMP family, is involved in the degradation of extracellular matrix (ECM) components. The natural inhibitors of MMP are TIMPs, and their mutual balance is crucial in maintaining the integrity of the ECM [71]. Since MMP-3 is involved in different physiological and pathological processes, including accumulation of inflammatory cells, stimulation of neo-angiogenesis, and supporting osteoclast differentiation simultaneously with inhibition of osteoblastogenesis and MSC proliferation, PoPEx could support all these mechanisms in P-GMSCs which are relevant for the inflammation control in chronic periodontitis.

The effect of LPS has been investigated on different types of dental MSCs, with the least research conducted on GMSCs [43,72]. Our results showed an inhibitory effect of LPS on the proliferation of both types of GMSCs. Even when applied in small concentrations, LPS significantly stimulates the expression of genes involved in inflammatory processes (*IL-6*, *IL-8*, *GRO-α*, *COX-2*), immunomodulation (*IDO-1*, *RANTES*, *TLR3*), and extracellular matrix degradation (*MMP-3*, *MMP-3/TIMP2* ratio). The expression of certain genes was differently modulated in the cell lines: increased expression of *MCP-1*, *TLR2*, *IGFB4*, and *HGF* genes only in P-GMSCs; inhibited expression of *TLR4*, *SDF1*, and *TIMP2*, and increased expression of *NFKB1* genes only in H-GMSCs. These findings suggest that LPS stimulates the proinflammatory functions of GMSCs, manifested by upregulation of proinflammatory cytokines/chemokines, similarly as demonstrated in other publications [73,74,75,76]. However, the modulatory potential of LPS also depended on whether GMSCs were isolated/established from healthy or inflamed gingiva.

Increased expression of *NFKB1* and *COX-2* aligns with the findings of cytokines/chemokines genes. *NFKB1* encodes a 105 kD Rel protein-specific transcription inhibitor, which is processed by a proteasome to produce a 50 kD protein, a DNA binding subunit of the NF-kB protein complex. NF-kB is a transcription regulator that is activated by different signals originating from intra- and extra-cellular sources, ROS, bacterial or viral products, and many other stimuli. Upon activation, NF-kB is translocated into the nucleus, where it stimulates the expression of genes involved in different biological functions, including the production of proinflammatory cytokines/chemokines and cytokines involved in the immune response [71,77]. COX-2, a key enzyme mediating prostaglandin synthesis (PGE2) and its gene, is an early response gene for many proinflammatory stimuli [78].

Treatment of MSCs with LPS was followed by increased expression of some genes involved in their immunosuppressive functions, such as *IDO-1*, *TLR3*, and *RANTES*. As already mentioned, signaling through TLR3 stimulates differentiation of immunosuppressive MSC2 type, and the process was additionally enhanced by LPS-induced downregulation of TLR4 [54], similarly as we obtained in our study. An increased COX-2 gene expression by LPS and subsequent PGE2 production contributes to the immunosuppressive capability of GMSCs. Namely, it is known that MSC-produced PGE2 exerts numerous immunosuppressive effects on dendritic cells, macrophages, and T cells while maintaining the basic functions of MSCs, such as proliferation, migration, and differentiation [79]. Therefore, LPS preserves/increases the immunosuppressive properties of GMSCs which is opposite to its proinflammatory role. In addition, LPS can stimulate extracellular matrix degradation by increasing MMP-3 mRNA expression and MMP-3/TIMP2 ratio in GMSCs. This finding is in line with the fact that bacterial endotoxin is an important factor in the pathogenesis of chronic periodontitis [80]. 

Regarding the impact of LPS on MSCs’ osteoblastogenesis, it has been observed that higher concentrations of LPS have an inhibitory effect on the expression of numerous genes involved in these processes, such as *RUNX2*, *BMP2*, *Alkaline Phosphatase* (*ALP*), *COL1A1*, and others. On the other hand, lower concentrations (less than 1 µg/mL) have a stimulatory effect [43,81]. We used lower LPS concentrations in our study because they are more physiologically relevant for modeling the course of chronic periodontitis. Since our aim was not to study osteoblastogenesis, a long culture process, we concentrated on early osteoblastic genes (*RUNX2*, *BMP2*, and *COL1A1*). Our results showed that PoPEx had no significant effect on the expression of these genes in unstimulated GMSCs, except for an upregulation of *BMP2* in P-GMSCs. LPS increased the expression of *RUNX2* in P-GMSCs and decreased the expression of *BMP2* in both lines. In addition, LPS decreased the expression of *COL1A1* and *OPG* in H-GMSCs. These results support the concept that H- and P-GMSCs had different basal characteristics, that even lower concentrations of LPS downregulated osteoblastogenesis in GMSCs from healthy gingiva and that LPS stimulated osteoblastogenesis in GMSCs established from inflamed gingiva. This conclusion is based on the findings that RUNX2 is the earliest osteogenic transcription factor [82]. Our study also showed that PoPEx significantly stimulates LPS-induced osteoblastogenesis in P-GMSCs by upregulating *RUNX2*, *BMP2*, and *COL1A1*, in contrast to H-GMSCs, where such an effect was not visible. In addition, OPG was upregulated. It is known that OPG is a soluble decoy receptor for the receptor activator of nuclear factor-kappa B ligand (RANKL). By blocking RANKL, OPG inhibits osteoclast formation and activity and stimulates osteoblast differentiation [83].

This was not the only key difference in the effect of PoPEx on the LPS-treated GMSCs between the H and P lines. PoPEx potentiated immunosuppressive properties of LPS-stimulated P-GMSCs, as judged by the upregulation of *IDO*, *TGF-β*, *TLR3*, and *COX-2* genes compared to LPS treatment alone. However, in H-GMSCs, higher concentrations of PoPEx inhibited *IDO-1*, *TGF-β*, and *COX-2* gene expression. Although explaining these differences requires additional investigation, it can be assumed that PoPEx can enhance the immune response in the early stage of periodontitis (model of LPS treatment of H-GMSCs). In contrast, during the exacerbation of chronic periodontitis (model of LPS treatment of P-GMSCs), PoPEx enhanced the immunosuppressive mechanisms, which could be beneficial for the resolution of inflammation and restriction of overactivated immune responses. Although PoPEx increased the expression of both *MMP-3* and *TIMP-2* in LPS-stimulated P-GMSCs, the *MMP-3/TIMP-2* ratio was lower than the index in only LPS-stimulated P-GMSC cultures, suggesting that PoPEx tends to restrict tissue destruction in chronic periodontitis. At the same time, stimulation of genes encoding proinflammatory cytokines and chemokines, including *IP-10* and *RANTES*, allows the unhindered development or even enhancement of host defense mechanisms at both stages of periodontitis. The story regarding other genes such as *IGFB4*, *HGF*, and *HIF-1α*, which were upregulated in LPS-stimulated P-GMSCs and were differently modulated in LPS-stimulated H-GMSCs is more complicated and needs additional investigations, bearing in mind the complexity of functions that the genes control.

In conclusion, our results suggest that PoPEx differently modulated the expression of genes in GMSCs under basal conditions and inflammatory microenvironment, mimicked by the treatment of GMSCs by LPS. The differences also existed whether GMSCs were established from healthy or periodontitis-affected gingival tissues. PoPEx treatment of unstimulated H-GMSCs was followed by inhibition of most genes associated with inflammation and enhancement of genes involved in immunosuppression. In contrast, PoPEx treatment was the most effective in the upregulation of genes for proinflammatory cytokines and chemokines, additional upregulation of genes associated with the suppression of the immune responses, genes involved in tissue regeneration and repair, and early genes involved in the stimulation of osteogenesis in LPS-stimulated P-GMSCs. The study has limitations in that gene expression dynamics were not investigated, the gene expression was not correlated with its products, and other functions of PoPEx-treated GMSCs were not investigated especially their differentiation possibilities. However, these results are starting points to understand better the particular role of individual components from the extract, especially punicalagin and ellagic acid, which are mostly investigated in different biological systems [84,85] but little on MSCs. In addition, the obtained results may further elucidate the complex role of GMSCs in health and inflammatory diseases and better understand the possible application of PoPEx and its constituents, in the form of mouthwash or oral/gingival gel for preventing gingivitis and periodontitis and treating chronic periodontitis as an adjunct therapeutic modality.

## 4. Materials and Methods

### 4.1. Tissue Donors and General Study Design

This was a collaborative study conducted at the Medical Faculty University of Banja Luka, Bosnia and Herzegovina, Institute for the Application of Nuclear Energy Research (INEP), University of Belgrade, Serbia (Laboratory part of the study), and Clinic for Dentistry (Department for Oral Surgery), and Military Medical Academy (MMA), Belgrade, Serbia (Clinical part of the study). Gingival tissue samples were collected from three donors with chronic periodontitis and three donors with healthy gingiva after written informed consent was obtained from the donors. Clinically healthy gingival samples were collected from subjects (male) who had no history of periodontal disease and smoking, aged 28, 36, and 40 years, respectively.

The periodontitis group patients (male) were 38, 45, and 48 years old. Periodontitis was diagnosed according to the American Academy of Periodontology (AAP) Classification of Disease. Staging and gradation of the disease were performed according to the Consensus report of workgroup 2 of the 2017 World Workshop on the Classification of Periodontal and Peri-Implant Diseases and Conditions [1]. All patients were classified as stage II-based. They had clinical attachment loss (CAL) of 4 mm at the site of the most significant loss, and the mean maximum probing depth was 4.33 mm. All patients were classified as periodontitis grade A (slow rate of progression). The subjects from both groups were non-smokers. The chronic periodontitis patients had no diabetes, malignant diseases, or systemic autoimmune diseases and did not receive antibiotics, vitamin supplements, or immunosuppressive drugs for two months before tissue sampling. Other data from the medical history of study participants were not recorded. Periodontitis gingival specimens were obtained during the flap debridement procedure, whereas healthy gingival tissues were taken during tooth extraction for orthodontic purposes.

A part of the gingival tissue from the donors was subjected to classical pathohistological processing. The paraffin sections were stained with hematoxylin and eosin (H&E) (Sigma-Aldrich, Darmstadt, Germany) and analyzed under a light microscope (Olympus, Hamburg, Germany).

### 4.2. Establishment of GMSC Lines

GMSC lines were established by the procedure that we have already published [15]. Gingival tissues were washed in phosphate-buffered saline (PBS). After that, the epithelial cell layer was removed by scalpel, minced with fine scissors, and digested with collagenase type II (5 µg/mL) and DNAse (40 IU/mL) in serum-free α-MEM for two hours at 37 °C and 5% CO_2_ in a cell incubator. All components were from Sigma-Aldrich, Darmstadt, Germany. The softened tissue was then gently pressed through a 30 µm nylon mesh, rinsed with α-MEM medium, and centrifuged at 1800 rpm for 10 min. The cell suspension was placed in 24-well cell culture dishes (Sarstedt, Numbrecht, Germany) at a density of 2000 per cm^2^ and cultured in a complete MSC medium. The medium contained α-MEM, 10% fetal calf serum (FCS), 100 IU/mL penicillin, 50 μg/mL streptomycin, 2.5 μg/mL amphotericin B (all from Thermo Fisher Scientific, Dreieich, Germany), 1% sodium pyruvate, and 100 µM L-ascorbate-2-phosphate (both from Sigma-Aldrich, Darmstadt, Germany). After three days, non-adherent cells were removed by washing the wells with α-MEM medium, followed by replacing the complete culture medium twice a week. When cell layers reached confluency, the detachment from the plastic was performed by treating the cells with 0.02% trypsin/0.02% Na EDTA (Sigma-Aldrich, Darmstadt, Germany) in PBS. The harvested cells were plated in 6-well cell culture dishes at a 5000 per 1 cm^2^ density in the complete culture medium without amphotericin B. The cells, which are further classified as GMSCs, were used for the experiments after the 4th passage.

### 4.3. In Vitro Differentiation of GMSCs

To determine osteogenic differentiation (OD), H- and P-GMSCs were plated at a density of 6 × 10^4^ cells on plastic coverslips inserted into six-well plates until they reached confluence. Then, GMSCs were cultured for 21 days in the complete α-MEM culture medium supplemented with 10% 10 nM dexamethasone (Galenika, Belgrade, Serbia), 10 mM glycerophosphate and 0.05 mM ascorbic acid (both from Sigma-Aldrich, Darmstadt, Germany). The osteogenic medium was changed twice a week. At the end of the cultivation period, coverslips were washed with PBS, fixed with 4% paraformaldehyde for 60 min at room temperature, washed twice with distilled water, and stained with 2% Alizarin Red (Sigma-Aldrich, Darmstadt, Germany) for 45 min. Finally, the coverslips were mounted on microscopic slides.

To induce adipogenic differentiation (AD), confluent monolayers of GMSCs were cultured for 21 days in the complete α-MEM medium supplemented with 0.5 µM dexamethasone (Galenika, Belgrade, Serbia), 0.5 µM isobutyl-methylxanthine (IBMX), (Sigma-Aldrich, Darmstadt, Germany) and with 50 µM indomethacin (R&D Systems, Minneapolis, MN, USA). The complete cultivation procedure was the same as that described for OD. Coverslips were washed with 60% isopropanol for 5 min and stained with 0.3% Oil Red O (Sigma-Aldrich, Darmstadt, Germany). At the end, the coverslips were washed with tap water stained with hematoxylin for 1 min and mounted on microscopic slides.

For chondrogenic differentiation (CD), GMSCs (5 × 10^5^) were placed in Eppendorf tubes and pelleted in by centrifugation (1800 rpm) for 10 min. The cell pellet was cultivated in the complete α-MEM medium supplemented with TGF-β3 (10 ng/mL) (R&D Systems, Minneapolis, MN, USA), dexamethasone (100 nM) (Galenika, Belgrade, Serbia), and ascorbic acid (50 ng/mL) (Sigma-Aldrich, Darmstadt, Germany) for 21 days. After cultivation, the cell pellets were cryopreserved in an embedding medium (Bio-Optica, Milan, Italy), frozen at −80 °C. Cryostat sections (Leica Biosystems, Barcelona, Spain) were air-dried and fixed with 4% paraformaldehyde, and stained with Alcian blue (Sigma-Aldrich, Darmstadt, Germany). The stained sections were washed with distilled water and counterstained with 0.1% Nuclear Fast Red solution (Sigma-Aldrich, Darmstadt, Germany). Negative controls for all differentiation procedures were GMSCs cultured in the complete basal α-MEM medium.

The stained cells/sections were observed under a light optical microscope (Olympus, Hamburg, Germany). All images were analyzed offline in ImageJ 1.47u software (National Institutes of Health, Bethesda, MD, USA). Semiquantitative analysis was performed as follows: Index 0—no visible positivity; Index 1—mild positivity of individual cells and the presence of 1–2 smaller mineralized islets (OD) or 1–2 positive cells (AD and CD) in at least 1 of the 10 analyzed microscopic fields; Index 2—mild positivity of individual cells and the presence of up to 5 small and medium-sized mineralized islets (OD) or up to 5 positive cells (AD and CD) in at least 2 of the 10 analyzed microscopic fields; Index 3—presence of up to 10 mineralized islets of different sizes (OD) or up to 10 positive cells (AD and CD) in at least 5 of the 10 analyzed microscopic fields; Index 4—presence of mineralized nuclei of all sizes, some merged (OD) or most positive cells (AD and CD) on all 10 analyzed microscopic fields.

### 4.4. Flow Cytometry

Phenotypic analysis of H-GMSCs and P-GMSCs was performed using monoclonal antibodies (mAbs) and flow cytometry. The cells were stained by the following mAbs using dilutions recommended by the manufacturer. Anti-CD14-FITC (63D3), anti-STRO-1-FITC (STRO-1), anti-CD45-APC (HI30), anti-CD90-PE (5E10), anti-CD73-biotin (AD2), anti-SSEA-4-biotin (MC-813-70), anti-CD166-PE (3A6), purified anti-NG2 (MEL62), and anti-CD105-APC (43A3) were obtained from all from BioLegend, Basel, Switzerland. Anti-PDGFRβ-Alexa Fluor 546 (D-6), anti-CD146-Alexa Fluor 488 (P1H12) (both from Santa Cruz Biotechnology, Dallas, TX, USA), anti-CD39-FITC (eBioA1) (eBioscience, San Diego, CA, USA), anti-CD34-FITC (581) (Elabscience, Wuhan, China), and anti-CD44-APCCy7 (IM7) were obtained from Sigma-Aldrich, Darmstadt, Germany. Anti-CD45-APC (HI30), anti-CD19-FITC (HIB19), and anti-CD3-PE (UCHT1) were from BioLegend, Basel, Switzerland. Streptavidin-APC and streptavidin APCCy7 were from BioLegend, Basel, Switzerland, whereas anti-mouse IgG and rabbit anti-goat polyclonal antibody-Alexa Fluor 488 were from Sigma-Aldrich, Darmstadt, Germany, and Abcam, Cambridge, UK, respectively. Negative isotype controls (mAbs conjugated with corresponding fluorochromes) were purchased from BioLegend, Basel, Switzerland. The mAbs were diluted in 2% FCS/0.01% NaN_3_ in PBS and incubated with the cells for 30 min at 4 °C. The labeled cells were analyzed on a BD LSR II flow cytometer (BD Biosciences, Franklin Lakes, NJ, USA). The doublets were excluded according to forward scatter (FSC)-A/FSC-H. More than 5000 gated cells were analyzed according to their specific FSC-A/side-scatter (SSC)-A properties. Signal overlaps between the channels were compensated before each experiment using single labeled cells, and non-specific fluorescence was determined by using the appropriate isotype controls. The acquired data were analyzed offline in the FlowJoVX program (BD Biosciences, Franklin Lakes, NJ, USA).

### 4.5. Immunohistochemistry and Confocal Microscopy

The gingival tissue was sectioned using a cryostat (Leica CM 1950, Wetzlar, Germany). Tissue sections (approximately 6 µm thick) were incubated with anti-CD146 mAb followed by peroxidase-conjugated anti-rabbit Ig. Both Ab were purchased from Abcam, Cambridge, UK. The reaction was visualized using diaminobenzidine (Sigma-Aldrich, Darmstadt, Germany). Immunohistochemical analysis was conducted using a light microscope (Olympus, Hamburg, Germany). For further identification of CD146+ cells, gingival tissue sections were stained with anti-CD146 AlexaFluor 488 (Santa Cruz Biotechnology, Dallas, TX, USA) and anti-CD31 AlexaFluor 433 (ThermoFisher Scientific, Dreieich, Germany) mAbs, followed by Syto59 nuclear stain (ThermoFisher). Analysis was performed using a confocal microscope (Zeiss LSM 510/Axiovert 200 M, Jena, Germany). DAPI (4′,6-diamidino-2-phenylindole) (ThermoFisher Scientific, Dreieich, Germany) was used to counterstain nuclei.

### 4.6. Preparation and Analysis of PoPEx

The detailed procedure for preparing and analyzing PoPEx was published in our previous paper [27]. The powdered pomegranate peel was prepared from pomegranate fruits collected at a natural locality in southern Bosnia and Herzegovina. The peel was extracted with 50% ethanol, using 1:10 as a solid-to-solvent ratio. After filtration and evaporation, the extract was analyzed spectrophotometrically by the Folin–Ciocalteu method, where gallic acid was used to prepare the calibration curve. The results were expressed as mg of gallic acid equivalents per gram of dry weight. The pomegranate peel was deposited in the Botanical Garden “Jevremovac” University of Belgrade (voucher specimen No. BEOU 17742). HPLC analysis was performed on Agilent 1200 RR HPLC (Agilent, Waldbronn, Germany), equipped with a DAD detector, using reverse-phase analytical column Zorbax SB-C18 (Agilent, Waldbronn, Germany). Detection was performed at 260 and 320 nm. The quantity of analyzed compounds (punicalagin, punicalin, gallic acid, and ellagic acid) was calculated using calibration curves of authentic standards (Appendix A). The results are expressed as mg per gram of dry weight. Experiments were repeated three times. Their mean content was as follows: punicalagin 67.26 ± 0.81 mg/g, punicalin 31.91 ± 0.22 mg/g, ellagic acid 25.11 ± 0.06 mg/g, and gallic acid 9.75 ± 0.05 mg/g.

### 4.7. MTT and Proliferation Assays

GMSCs were cultivated in 96-well plates (5 × 10^4^/well; triplicates), in either the complete α-MEM medium or the medium with different dilutions of PoPEx. After a 48 h incubation period, the solution of 3-[4,5-dimethyl-2-thiazolyl]-2,5-diphenyl tetrazolium bromide (MTT) (Sigma-Aldrich) (100 μL/well, final concentration 100 μg/mL), was added to the wells. Wells containing only different concentrations of PoPEx in the complete α-MEM medium were used to test the interaction of MTT-developed color with the extract. The wells with MTT served as blank controls. The plates were incubated with MTT for 4 h in an incubator at 37 °C, and after that, the formazan crystals were dissolved with 0.1N HCl/10% SDS (sodium dodecyl sulfate) (100 μL/well) overnight. The developed color’s optical density (OD) was read at 570/650 nm (ELISA reader, Behring II, Marburg, Germany). The results were presented as the relative metabolic activity in PoPEx-treated cultures compared to the metabolic activity of control cultures without PoPEx, where OD was used as 100%. The relative metabolic activity was calculated as follows: metabolic activity (%) = (OD of cultures with PoPEx − OD with PoPEx without cells/OD of control cultures without PoPEx − OD of medium without cells) × 100.

In the proliferation assay, GMSCs (2.5 × 10^4^/well; triplicates) were incubated with PoPEx (two different concentrations) with or without LPS (50 ng/mL) (Sigma) in the complete α-MEM medium for 3 days. Control cultures were GMSCs without PoPEx. After cultivation, cells were treated with 0.25% trypsin to detach the cells from the plastic. The released cells were pelleted by centrifugation, and after that, the number of cells in each well was calculated by a cytometer. Trypan blue was used to detect cell viability which was about 98%, in each well. The relative proliferation of cells (cell growth) was expressed based on the number of recovered cells compared to the number of cells in control cultures, used as 100%. The calculation was as follows: relative proliferation (%) = number of cells in experimental cultures/number of cells in control cultures × 100.

### 4.8. Real-Time Quantitative Polymerase Chain Reaction

GMSCs were cultivated in plastic flasks (bottom square 25 cm^2^) (Sarsted, Dordreht, Germany) until confluence (usually for two days). After that, each flask was treated with 10 µg/mL or 40 µg/mL of PoPEx, with or without LPS (50 ng/mL). GMSCs cultivated alone served as controls. After 24 h, the cells were detached from the plastic substrate with 0.25% trypsin, pelleted by centrifugation, and stored in Trizol reagent (Thermo Fisher Scientific, Dreieich, Germany) at −80 °C until analysis. Total RNA was extracted from cultured cells using the Total RNA Purification Mini Spin Kit (Genaxxon Bioscience GmbH, Ulm, Germany) following the manufacturer’s protocol. A high-capacity cDNA reverse transcription kit (Thermo Fisher Scientific, Dreieich, Germany) was used to transcribe 0.1 µg of isolated RNA as a template. The synthesized cDNA was then subjected to Real-Time Quantitative Polymerase Chain Reaction (qPCR) using a SYBR Green PCR Master Mix (Thermo Fisher Scientific, Dreieich, Germany) in a 7500 real-time PCR machine (Applied Biosystems, Waltham, MA, USA). The conditions were: 10 min at 95 °C activation, 40 cycles of 15 s at 95 °C and 60 s at 60 °C. The results were normalized against β-actin for each sample and expressed as a relative target abundance (versus the non-treated sample of each cell line) using the 2^−ΔΔCt^ method [86]. To compare differences in the expression of each marker between H-GMSCs and P-GMSCs upon PoPEx, LPS, or PoPEx-LPS treatment, mRNA expression of each marker was calculated for each cell line as a fold change of basic level expression used as 1. To compare variances in basic expression levels of analyzed markers between non-treated H-GMSCs and P-GMSCs, the expression of each marker on non-treated cells was calculated as fold change of marker expression in H-GMSCs of one donor used as 1. Primers used in the study are listed in Table 2. All primers were purchased from Thermo Fisher Scientific, Dreieich, Germany.

### 4.9. Statistics

To assess differences between H- and P-GMSCs parameters or between experimental and appropriate control samples Kruskal–Wallis or Mann–Whitney tests were used. Differences in mRNA expression between untreated, LPS-stimulated, PoPEx-treated, and LPS + PoPEx-treated GMSCs were analyzed using a ratio-paired *t*-test or Wilcoxon test. Values at *p* < 0.05 or less were considered to be statistically significant. The statistical analysis and graphs were performed in GraphPad Prism version 8.0.0 (GraphPad Software, San Diego, CA, USA).

## Figures and Tables

**Figure 1 ijms-24-15407-f001:**
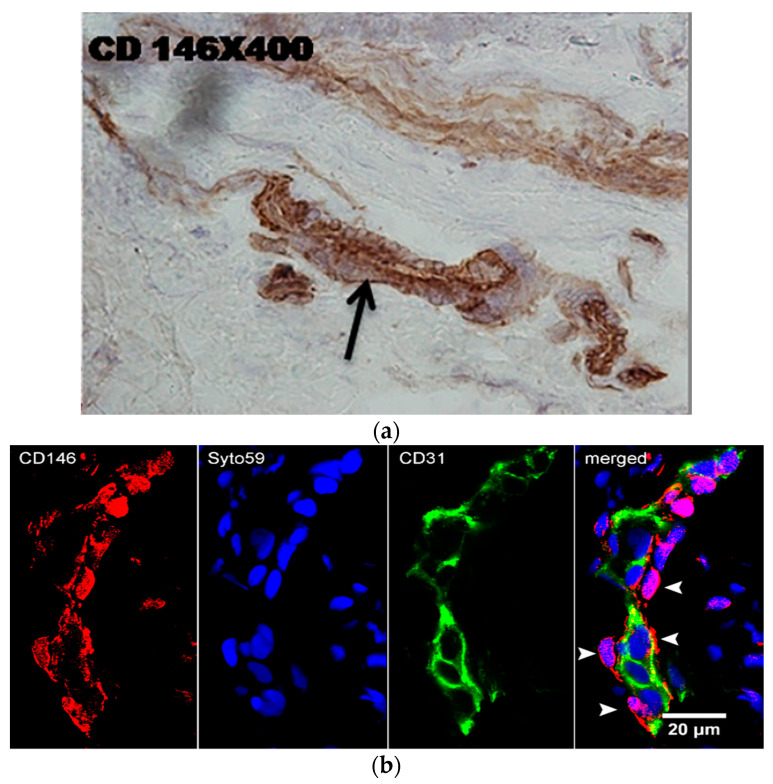
Immunostaining of gingival tissue sections with anti-CD146 mAb (immunoperoxidase) (**a**) and anti-CD146-AlexaFluor 466 (red) and anti-CD31-AlexaFluor 433 (green) processed for confocal microscopy (**b**). Nuclei are stained in blue by Syto59. Note single CD146+ pericytes (arrowheads).

**Figure 2 ijms-24-15407-f002:**
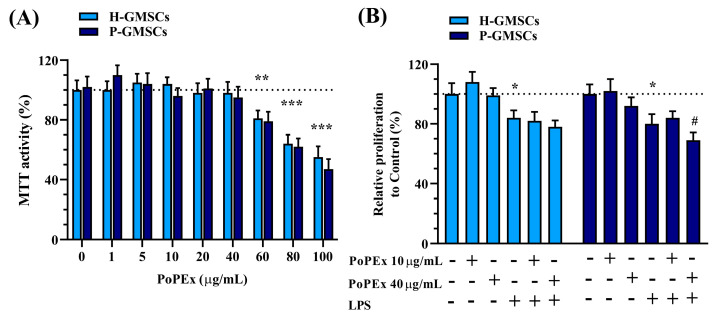
The effect of PoPEx on the viability (MTT activity) (**A**) and proliferation (**B**) on H- and P-GMSCs in culture. Values are given as mean ± SD (*n* = 3). * *p* < 0.05; ** *p* < 0.01; *** *p* < 0.005 compared to corresponding controls (non-treated H- or P-GMSCs). # *p* < 0.05 compared to LPS-stimulated P-GMSCs.

**Figure 3 ijms-24-15407-f003:**
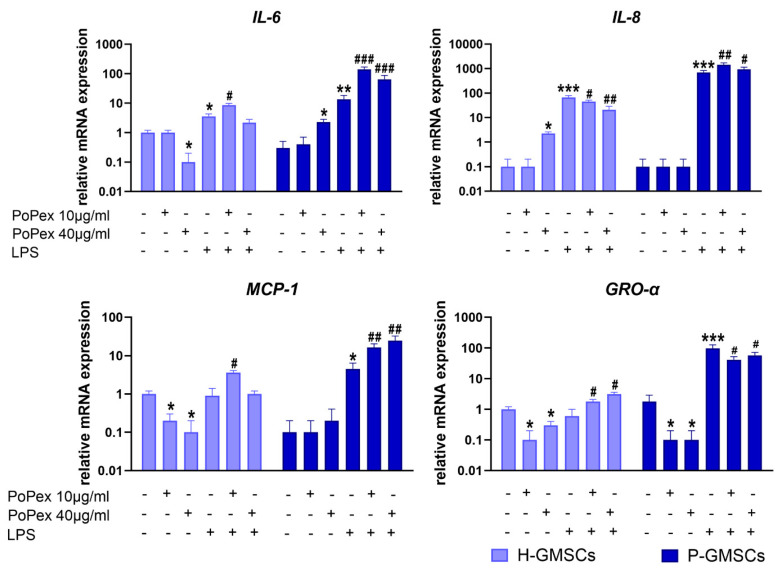
The effect of PoPEx on the expression of IL-6, IL-8, MCP-1, and GRO-α mRNA in H- and P-GMSCs. Values are given as mean ± SD (*n* = 3). * *p* < 0.05; ** *p* < 0.01; *** *p* < 0.005 compared to corresponding controls (non-treated H- or P-GMSCs). # *p* < 0.05; ## *p* < 0.01; ### *p* < 0.005 compared to corresponding LPS-stimulated GMSCs.

**Figure 4 ijms-24-15407-f004:**
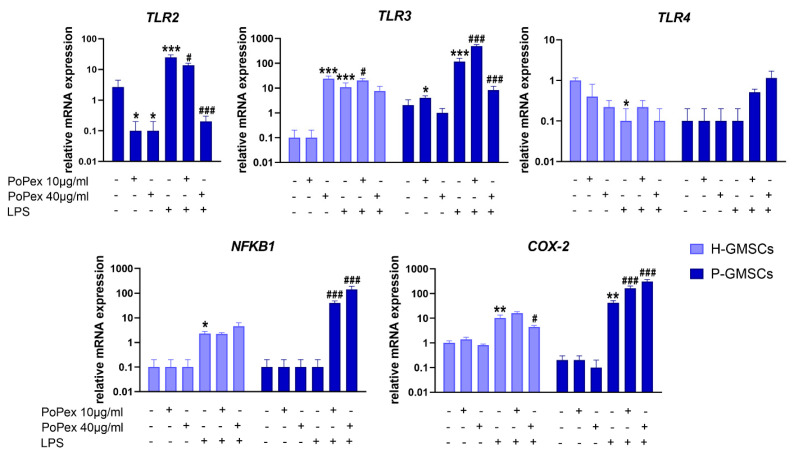
The effect of PoPEx on the expression of TLR2, TLR3, TLR4, NFKB1, and COX-2 mRNA in H- and P-GMSCs. Values are given as mean ± SD (*n* = 3). * *p* < 0.05; ** *p* < 0.01; *** *p* < 0.005 compared to corresponding controls (non-treated H- or P-GMSCs). # *p* < 0.05; ### *p* < 0.005 compared to corresponding LPS-stimulated GMSCs.

**Figure 5 ijms-24-15407-f005:**
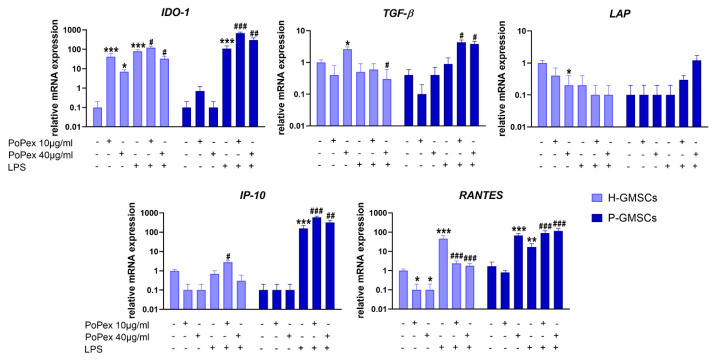
The effect of PoPEx on the expression of IDO-1, TGF-β, LAP, IP-10, and RANTES mRNA in H- and P-GMSCs. Values are given as mean ± SD (*n* = 3). * *p* < 0.05; ** *p* < 0.01; *** *p* < 0.005 compared to corresponding controls (non-treated H- or P-GMSCs). # *p* < 0.05; ## *p* < 0.01; ### *p* < 0.005 compared to corresponding LPS-stimulated GMSCs.

**Figure 6 ijms-24-15407-f006:**
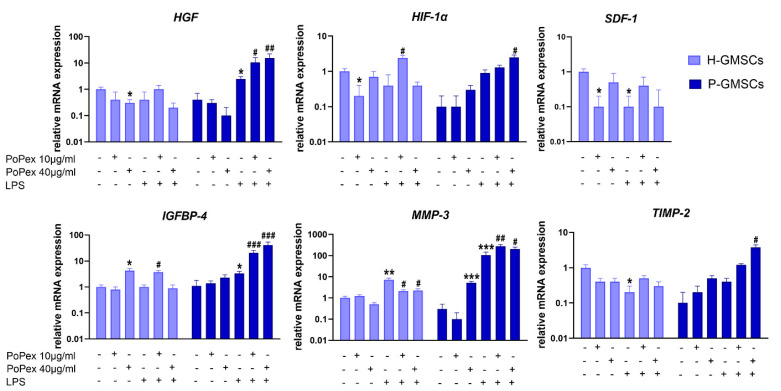
The effect of PoPEx on the expression of HGF, HIF-α, SDF-1, IGFB4, MMP-3, and TIMP-2 mRNA in H- and P-GMSCs. Values are given as mean ± SD (*n* = 3). * *p* < 0.05; ** *p* < 0.01; *** *p* < 0.005 compared to corresponding controls (non-treated H- or P-GMSCs). # *p* < 0.05; ## *p* < 0.01; ### *p* < 0.005 compared to corresponding LPS-stimulated GMSCs.

**Figure 7 ijms-24-15407-f007:**
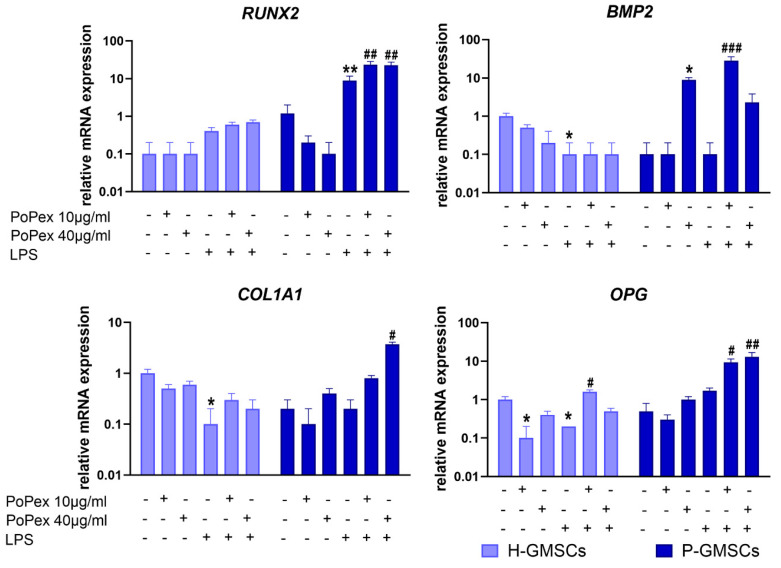
The effect of PoPEx on the expression of RUNX2, BMP2, COL1A1, and OPG mRNA in H- and P-GMSCs. Values are given as mean ± SD (*n* = 3). * *p* < 0.05; ** *p* < 0.01; compared to corresponding controls (non-treated H- or P-GMSCs). # *p* < 0.05; ## *p* < 0.01; ### *p* < 0.005 compared to corresponding LPS-stimulated GMSCs.

**Table 1 ijms-24-15407-t001:** Phenotypic characteristics of GMSC lines and their differentiation capabilities.

	H—GMSCs	P—GMSCs
Phenotype	%	%
CD90	95.2 ± 2.2	94.6 ± 2.0
CD166	96.1 ± 1.8	95.6 ± 2.1
CD73	97.7 ± 1.2	98.0 ± 0.8
CD39	96.6 ± 1.4	97.7 ± 1.3
CD44	94.6 ± 2.1	95.6 ± 1.6
CD105	99.2 ± 0.3	98.1 ± 1.0
CD146	32.4 ± 10.2	43.6 ± 8.7
PDGF—R	11.3 ± 4.2	14.6 ± 3.8
SSEA—4	9.9 ± 4.8	15.5 ± 3.3
STRO—1	12.6 ± 4.0	12.9 ± 3.6
NG—2	11.6 ± 3.4	16.6 ± 5.4
CD34	1.6 ± 0.2	1.1 ± 0.2
CD45	1.2 ± 0.4	1.6 ± 0.2
CD14	0.8 ± 0.2	1.1 ± 0.3
CD3/CD19	0.9 ± 0.2	0.9 ± 0.2
Differentiation	Index	Index
Osteogenesis	2.6 ± 0.8	4.1 ± 0.3 *
Chondrogenesis	1.8 ± 0.4	2.1 ± 0.4
Adipogenesis	0.9 ± 0.5	0.6 ± 0.2

Values are given as mean ± SD (*n* = 3); * *p* < 0.05 compared to H-GMSCs.

**Table 2 ijms-24-15407-t002:** Sequences of the primer pairs used for the real-time PCR experiments.

Primers	Sequence
*h HGF forward* *h HGF reverse*	GCACTGACTCCGAACAGGAT CAGGAGTTTGGTCACCCACA
*h MCP-1 forward* *h MCP-1 reverse*	GATCTCAGTGCAGAGGCTCG TTTGCTTGTCCAGGTGGTCC
*h HIF-1α forward* *h HIF-1α reverse*	GTCTGAGGGGACAGGAGGAT CTCCTCAGGTGGCTTGTCAG
*h IGFBP4 forward* *h IGFBP4 reverse*	TCTGAGCCCTGGTGTGTTTC GCTGGCACGTAGTACATGGT
*h GRO-α forward* *h GRO-α reverse*	CTGGCTTAGAACAAAGGGGCT TAAAGGTAGCCCTTGTTTCCCC
*h LAP forward* *h LAP reverse*	ACTGCCCAGTTCAAGAGACG CCGACCGGATCTGTACTTCG
*h RANTES forw.* *h RANTES reverse*	CAGTCGTCTTTGTCACCCGA CGGGTGGGGTAGGATAGTGA
*h OPG forward* *h OPG reverse*	TAACGTGATGAGCGTACGGG GCAGCACAGCAACTTGTTCA
*h SDF-1 forward* *h SDF-1 reverse*	GGACTTTCCGCTAGACCCAC GTCCTCATGGTTAAGGCCCC
*h IDO-1 forward* *h IDO-1 reverse*	GGGAAGCTTATGACGCCTGT CTGGCTTGCAGGAATCAGGA
*h TLR3 forward* *h TLR3 reverse*	CCTTTTGCCCTTTGGGATGC TGAAGTTGGCGGCTGGTAAT
*h TLR2 forward* *h TLR2 reverse*	TGAGCTGCCCTTGCAGATAC TGCAAGCAGGATCCAAAGGA
*h TLR4 forward* *h TLR4 reverse*	GGATTTCACACCTCCACGCA GGTCAGAGCGTGATAGCGAG
*h MMP-3 forward* *h MMP-3 reverse*	TGAAATTGGCCACTCCCTGG GGAACCGAGTCAGGTCTGTG
*h TIMP-2 forward* *h TIMP-2 reverse*	TCTCGACATCGAGGACCCAT TGGACCAGTCGAAACCCTTG
*h TGF-β forward* *h TGF-β reverse*	CCGGGTTATGCTGGTTGTACAG AAGGACCTCGGCTGGAAGTGG
*h IL-6 forward* *h IL-6 reverse*	CACTCACCTCTTCAGAACGA CTGTTCTGGAGGTACTCTAGG
*h IL-8 forward* *h IL-8 reverse*	ACACAGAGCTGCAGAAATCAGG GGCACAAACTTTCAGAGACAG
*h IP-10 forward*	AGCAGAGGAACCTCCAGTCT
*h IP-10 reverse*	ATGCAGGTACAGCGTACAGT

Values are given as mean ± SD (*n* = 3); *p* < 0.05 compared to H-GMSCs.

## Data Availability

All data are included in this article.

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
