# Peer review of "Pomegranate Peel Extract Differently Modulates Gene Expression in Gingiva-Derived Mesenchymal Stromal Cells under Physiological and Inflammatory Conditions"

_ijms, 2023, doi:10.3390/ijms242015407_

Round 1

Reviewer 1 Report

The work is interesting and addresses a topic felt in the dental field.

The work was well conducted and the results are clear and detailed.

The part that should be improved in my opinion are the conclusions.

Some indications could be given for the use of pomegranate extract for clinical applications to prevent and combat gingivitis and periodontitis. A comparison with the already known calendula and aloe extracts would be very interesting.

Author Response

Thank you so much for the objective review. The authors accepted the suggestions, and possible indications for clinical application are added at the end of the discussion chapter. In the Introduction, two articles are cited relating to the use of Callendulla officinalis and Aloe vera extracts in the treatment of gingivitis/periodontitis.

What has been added is highlighted in red

Reviewer 2 Report

This is very fresh and interesting study. However I found mostly minor flaws:

1. Introduction:

-Authors have to clarify why this study is novel and important. Please rationale the study before the aim of study.

- Please define clear aim of the study at the end of Introduction.

- Authors did not write anything about other natural products which can be use in periodontitis prevention, chronic and acute periodontitis treatment and diagnostics as well as other gingiva conditions management. Please add a paragraph about it. Authors can use the following highly reliable and the latest literature (use of the mentioned literature is voluntarily):

Rahman S, Nagur Karibasappa S, Mehta DS. Evaluation of the wound-healing potential of the kiwifruit extract by assessing its effects on human gingival fibroblasts and angiogenesis. Dent Med Probl. 2023;60(1):71–77. doi:10.17219/dmp/146635

Paradowska-Stolarz A, Wieckiewicz M, Owczarek A, Wezgowiec J. Natural Polymers for the Maintenance of Oral Health: Review of Recent Advances and Perspectives. Int J Mol Sci. 2021 Sep 25;22(19):10337. doi: 10.3390/ijms221910337.

Cieszkowski J, Warzecha Z, Ceranowicz P, Ceranowicz D, Kusnierz-Cabala B, Pedziwiatr M, Dembinski M, Ambrozy T, Kaczmarzyk T, Pihut M, Wieckiewicz M, Olszanecki R, Dembinski A. Therapeutic effect of exogenous ghrelin in the healing of gingival ulcers is mediated by the release of endogenous growth hormone and insulin-like growth factor-1. J Physiol Pharmacol. 2017 Aug;68(4):609-617.

Mazur M, Ndokaj A, Bietolini S, Nisii V, Duś-Ilnicka I, Ottolenghi L. Green dentistry: Organic toothpaste formulations. A literature review. Dent Med Probl. 2022;59(3):461–474. doi:10.17219/dmp/146133

2. Please describe limitations of the study at the end of Discussion. Authors have to also add a paragraph which describe potential clinical utility of obtained findings.

3. As I wrote before this is very good and well reported study. However for potential readers is difficult to read whole paper because is long. Therefore I recommend to prepare professional graphical abstract and add a figure/table which presents the most important findings from the study.

The manuscript needs gentle language revision after all corrections.

Author Response

-Authors have to clarify why this study is novel and important. Please rationale the study before the aim of study.

-Please define clear aim of the study at the end of Introduction. 

Reply: This part is rewritten in the Introduction (see lines 117-136)

- Authors did not write anything about other natural products which can be use in periodontitis prevention, chronic and acute periodontitis treatment and diagnostics as well as other gingiva conditions management. Please add a paragraph about it. Authors can use the following highly reliable and the latest literature (use of the mentioned literature is voluntarily):

Reply: We accepted the suggestion and added several sentences on this topic. Of the suggested references, we used two (two were of less importance) and also added four new ones. Two of the references were related to Calendula officinalis and Aloe vera, as suggested by the first reviewer. We tried to be as concise as possible bearing in mind the content of our paper (see lines 88-99).

-Please describe limitations of the study at the end of Discussion. Authors have to also add a paragraph which describe potential clinical utility of obtained findings.

Reply: The limitations of the study and potential clinical utility have already been written in the original manuscript. We have made some extensions now (lines 580-583; 588-590)

As I wrote before this is very good and well reported study. However for potential readers is difficult to read whole paper because is long. Therefore I recommend to prepare professional graphical abstract and add a figure/table which presents the most important findings from the study.

Reply: The idea about the graphical abstract is good. We drew it (see attachment). We think that the graphical abstract is sufficiently informative and that there is no need to duplicate it with another graph or table.

The manuscript needs gentle language revision after all corrections.

Reply: The language has been checked again and some minor corrections are marked in red
